# Assessment of Acute and Short-Term Developmental Toxicity of Mercury Chloride to Rare Minnow (*Gobiocypris rarus*)



Xiaoqin Xiong [1,2], Qingchao Shi [1,2], Hao Liu [1], Qian Zhou [1], Huatao Li [1], Peng Hu [1,2], Zhengyong Wen [1,2], Jianwei Wang [3], Yuanchao Zou [1,2], Yu Zeng [4,*,†] and Yaotong Hao [5,*,†]

1    College of Life Science, Neijiang Normal University, Neijiang 641112, China
2    Key Laboratory of Conservation and Utilization of Fish Resources, College of Life Sciences, Neijiang Normal University, Neijiang 641100, China
3    Institute of Hydrobiology, Chinese Academy of Sciences, Wuhan 430072, China
4    College of Life Science, China West Normal University, Nanchong 637000, China
5    Ocean College, Hebei Agricultural University, Qinhuangdao 066003, China
*    Correspondence: fish2021yes@163.com (Y.Z.); haoyaotong@163.com (Y.H.)
†    These authors contributed equally to this work.

**Abstract:** Mercury (Hg), as the most potentially hazardous heavy metal, has accumulated in the aquatic environment and has caused concern about its safety. To test the toxic effects of mercury chloride ($Hg^{2+}$) on rare minnow (*Gobiocypris rarus*), the acute toxicity of $Hg^{2+}$ to embryos, newly hatched larvae, juvenile fish, and the short-term developmental toxicity of $Hg^{2+}$ to the embryo and sac-fry stages, were investigated. The 96-h $LC_{50}$ values of $Hg^{2+}$ to embryos, newly hatched larvae, and juvenile fish were 0.56, 0.07, and 0.10 mg/L, respectively, suggesting newly hatched larvae were the most sensitive, followed by juvenile fish, while embryos were the most resistant in response to an $Hg^{2+}$ challenge. The research data revealed that the safe level of $Hg^{2+}$ exposure for rare minnow was 0.7 μg/L. In addition, the LOEC (lowest observed effect concentration) and NOEC (no observed effect concentration) values of $Hg^{2+}$ to heartbeat, mortality, malformation rate, and body length of survived larvae were identically, 0.05 and 0.01 mg/L, respectively. These findings indicated that $Hg^{2+}$ had lethal effects on rare minnow at different life stages, and that newly hatched larvae were the most sensitive stage. The above findings have important implications for better understanding the environmental risk assessment of $Hg^{2+}$ on aquatic organisms.

**Keywords:** acute toxicity; developmental toxicity; embryos; $Hg^{2+}$; rare minnow

## 1. Introduction

Mercury (Hg), as a potentially hazardous heavy metal [1], poses a great threat to organisms and to human health [2]. The toxicity of Hg pollution to humans and wildlife has received worldwide attention due to increasing Hg concentrations in recent years [3]. However, far less is known about the toxicity of Hg to fish [4]. The U.S. EPA has reviewed the water quality criteria (WQC), but related information is still limited in China. In addition, the WQC may vary in China and other countries due to differences in species groups and geography [5]. Moreover, the WQC are derived from toxicity tests on sensitive organisms, such as fish [6]. The key to these tests lies in the understanding that the most sensitive stage of the fish life history should be included [7]. Therefore, it is pressing to establish the WQC in China [8].

The toxicity of many environmental pollutants has been investigated, and biomarkers are widely used for ecotoxicological evaluation of water and sediment [9,10]. Aquatic ecosystems are the final ultimate sink for heavy metals including Hg. Hg concentrations vary from 0.038 to 10.6 μg/L in China [11]. It is well known that Hg exposure can produce neurotoxic, teratogenic effects, and genetic toxicity in aquatic organisms [12]. Among these organisms, fish are ideal models for evaluating the negative effects of Hg in the water [13].

Previous studies have shown that Hg is toxic to fish at different life stages [12], including embryonic [14], larval [15], juvenile [16], and adult stages [17]. Generally, fish at the larval stage are more sensitive to Hg than adult fish. However, fish embryos represent a variable relative sensitivity under different challenges. Praskova et al. [18] found that zebrafish (*Danio rerio*) embryos were more sensitive to diclofenac than juvenile fish. Mu et al. [19] compared the acute toxicity of difenoconazole to the three phases of zebrafish, and followed the order of larvae (1.17 mg/L) > adult fish (1.45 mg/L) > embryo (2.34 mg/L). On the other hand, Oliveira et al. [20] found similar 96 h $LC_{50}$ values of triclosan for zebrafish embryos and adults (0.42 and 0.34 mg/L, respectively). However, these results must be used cautiously, because they are easily affected by species-specific differences and various environmental conditions [4]. Furthermore, few studies focused on $LC_{50}$ for embryos, larvae, and juveniles of the same fish under comparable experimental conditions.

Rare minnow (*Gobiocypris rarus*), as a standardized aquatic test species (GB/T29763-2013) in China, have been widely used in environmental risk assessment of heavy metals [21]. Fish embryos have also been widely applied to toxicity testing owing to their advantages of animal welfare, low cost and high sensitivity. Thus far, the short-term toxicity test on rare minnow at embryo and sac-fry stages has been a standard method due to its high sensitivity to chemical testing [22]. Although the toxicity of mercury chloride ($Hg^{2+}$) to rare minnow larvae has been described by Li et al. [23], the developmental toxicity of Hg to rare minnow at embryonic-larval stages is still unclear.

In this study, the acute toxicity of $Hg^{2+}$ to embryos, newly hatched larvae, juvenile fish, and the short-term developmental toxicity of Hg to rare minnow at embryo and sac-fry stages, were examined. The aims of the study were to (1) compare the acute toxicity of $Hg^{2+}$ to rare minnow at different life stages, and (2) elucidate the developmental toxicity of $Hg^{2+}$ to rare minnow at embryo and sac-fry stages. Our findings will contribute to the aquatic environmental risk assessment of Hg and be helpful for a better understanding of the toxic mechanisms in rare minnow.

## 2. Material and Methods

### 2.1. Test Chemicals and Animals

Mercury chloride (analytical grade 99.5% purity) was obtained from the Shanghai Sinopharm Group Co., Ltd. (Shanghai, China). A 10 mg/L $Hg^{2+}$ stock solution was prepared in deionized water and stored at 4 °C until use in the tests. Rare minnow were provided by the Institute of Hydrobiology, the Chinese Academy of Sciences, and they were acclimated for at least two weeks before experiments. The fish cultural condition and egg collection conform to the established protocols [24].

### 2.2. Acute Toxicity Assays of $Hg^{2+}$ to Rare Minnow

The embryonic acute toxicity test used methods established by Zhu et al. [25]. Based on the preliminary experiments, six test concentrations (0.00, 0.20, 0.40, 0.60, and 0.80 mg/L $Hg^{2+}$) were prepared with aerated tap water, and the blank group (0.00 mg/L $Hg^{2+}$) was set as a control. We conducted three replicates at each concentration setting. Fertilized eggs approximately 8 h post-fertilization were randomly transferred to 24-well plates for 96 h exposure. Each well contained a 2 mL test solution and one embryo. All test 24-well plates were preserved in a biochemical incubator (25 ± 1 °C, 12:12 h light/dark photoperiod). The test solution was renewed daily, and dead individuals were removed immediately.

The larvae acute test was conducted following the operations described by Luo et al. [22] with a little modification. Newly hatched larvae per treatment were exposed to five levels of $Hg^{2+}$ (0.00, 0.08, 0.10, 0.12, 0.14 mg/L) with three replicates. Eighteen glass dishes were used, and each beaker contained a 200 mL test solution and ten larvae. The juvenile toxicity assay was performed according to OECD guideline No. 203. A total of 150 healthy juvenile fish, with a body weight of 0.25 ± 0.12 g and a body length of 25.00 ± 1.00 mm, were randomly distributed in 15 test beakers containing a 2 L test solution. Ten fish were exposed to each of five nominal experimental concentrations of $Hg^{2+}$ (0.00, 0.09, 0.10, 0.11

and 0.12 mg/L), based on the basis of preliminary experiment results. Both exposures lasted for 96 h. Fifty percent of the test solutions were replaced once daily to maintain the concentrations. During the experiment, larvae and juvenile fish were not fed and dead individuals were removed immediately from the beaker, and the mortality was recorded every day.

### 2.3. Short-Term Toxicity Test of $Hg^{2+}$ on Rare Minnow at Embryo and Sac-Fry Stages

The short-term toxicity test on rare minnow at embryo and sac-fry stages was conducted according to OECD guideline No. 212. Based on the embryonic acute toxicity test results, 20 embryos in each treatment were exposed to five levels of $Hg^{2+}$ (0.00, 0.001, 0.005, 0.01, 0.05, 0.10 mg/L) with three replicates at 8 hpf. About 300 embryos were randomly transferred into 24-well plates, with each well containing a 2 mL test solution and one embryo. No food was provided, and 50% of the exposure solutions were renewed once daily to maintain the test concentrations. Afterward, the spontaneous movement at 36 hpf, the heartbeat at 48 hpf, and the malformation were checked using an inverted dissecting microscope. The hatching rates at 72, 84, and 96 hpf were also examined. The body length of larvae after exposure was measured using an image processing and analysis system (Motic Images Plus 2.0, Motic, Xiamen, China). Dead individuals were recorded and removed from the wells during the experiment to preserve the water quality.

### 2.4. Chemical Analysis

Concentrations of $Hg^{2+}$ in all assays were determined by cold vapor atomic absorption spectrometry (CV-AAS) twice daily. Water temperature, pH, dissolved oxygen, and conductivity of the exposure solutions were determined daily using a water quality analysis system (Model HQ30d, Hach Inc., Loveland, CO, USA).

### 2.5. Statistical Analysis

Data about the mortality of fish and embryos at individual test concentrations were subjected to a probit analysis to determine the $LC_{50}$. All data were checked for assumptions of homogeneity of variance assumptions using Levene's test. When meeting assumptions, data were performed using the analysis of variance and then compared with the control values through the Waller–Duncan multiple-comparison test. The no-observed-effect concentration (NOEC) and the lowest-observed-effect concentration (LOEC) were determined based on the statistical results. All data were statistically analyzed using SPSS 19.0 software (SPSS Inc., Chicago, IL, USA). Values were expressed as mean ± standard deviation (SD). $p < 0.05$ was considered to be significant. All figures were drawn using Origin Pro 9.0 (OriginLab, Northampton, MA, USA) and Photoshop CS 7.0 (Adobe, San Jose, CA, USA).

## 3. Results

### 3.1. Physicochemical Parameters of the Test Solutions

Water temperature, pH and DO kept a relatively stable level during the test; the deviations between nominal and actual $Hg^{2+}$ concentrations were less than 20% in all tests (Table 1), indicating that the nominal concentration could reflect the actual content of $Hg^{2+}$ in this study according to OECD guidelines.

### 3.2. Acute Toxicity of $Hg^{2+}$ to Rare Minnow

The 24-h, 48-h, 72-h and 96-h $LC_{50}$ values of $Hg^{2+}$ to rare minnow are summarized in Table 2. The acute toxicity results showed that the 96-h $LC_{50}$ values of $Hg^{2+}$ to embryos, newly hatched larvae and juvenile fish were 0.56, 0.07, and 0.10 mg/L, respectively. This suggested that the order of lethal sensitivity of rare minnow at different life stages to $Hg^{2+}$ follows the order: newly hatched larvae > juvenile fish > embryos.

**Table 1.** Physicochemical parameters of water quality during the test.

| Test | Stages | Temperature (°C) | pH | DO (mg/L) | Nominal $Hg^{2+}$ (mg/L) | Measured $Hg^{2+}$ (mg/L) |
|---|---|---|---|---|---|---|
| Acute toxicity assays | Embryo | 24.84 ± 0.36 | 7.58 ± 0.36 | 7.52 ± 1.26 | Control | ND |
| | | 24.78 ± 0.42 | 7.62 ± 0.42 | 7.49 ± 1.64 | 0.20 | 0.18 ± 0.01 |
| | | 24.63 ± 0.55 | 7.68 ± 0.29 | 7.63 ± 1.79 | 0.40 | 0.37 ± 0.02 |
| | | 24.47 ± 0.54 | 7.69 ± 0.66 | 7.58 ± 1.26 | 0.60 | 0.56 ± 0.03 |
| | | 25.03 ± 0.32 | 7.70 ± 0.48 | 7.69 ± 1.39 | 0.80 | 0.75 ± 0.04 |
| | Newly hatched larvae | 24.62 ± 0.47 | 7.72 ± 0.42 | 7.74 ± 1.62 | Control | ND |
| | | 24.83 ± 0.36 | 7.63 ± 0.38 | 7.66 ± 1.37 | 0.08 | 0.07 ± 0.003 |
| | | 25.02 ± 0.28 | 7.58 ± 0.52 | 7.70 ± 1.68 | 0.10 | 0.09 ± 0.004 |
| | | 24.75 ± 0.37 | 7.66 ± 0.46 | 7.71 ± 1.46 | 0.12 | 0.10 ± 0.007 |
| | | 24.66 ± 0.58 | 7.57 ± 0.62 | 7.68 ± 1.08 | 0.14 | 0.12 ± 0.015 |
| | Juvenile fish | 25.03 ± 0.47 | 7.61 ± 0.41 | 7.72 ± 1.28 | Control | ND |
| | | 24.86 ± 0.39 | 7.58 ± 0.50 | 7.66 ± 1.48 | 0.09 | 0.084 ± 0.003 |
| | | 24.79 ± 0.46 | 7.62 ± 0.47 | 7.70 ± 1.38 | 0.10 | 0.092 ± 0.003 |
| | | 24.82 ± 0.53 | 7.60 ± 0.42 | 7.73 ± 1.06 | 0.11 | 0.101 ± 0.007 |
| | | 25.12 ± 0.37 | 7.56 ± 0.64 | 7.63 ± 1.20 | 0.12 | 0.105 ± 0.007 |
| Sub-chronic toxicity tests | Embryo and sac-fry | 25.03 ± 0.67 | 7.62 ± 0.39 | 7.64 ± 1.66 | Control | ND |
| | | 24.78 ± 0.53 | 7.59 ± 0.42 | 7.70 ± 1.78 | 0.001 | 0.0009 ± 0.0001 |
| | | 25.16 ± 0.48 | 7.60 ± 0.53 | 7.72 ± 1.16 | 0.005 | 0.0046 ± 0.0002 |
| | | 24.84 ± 0.39 | 7.58 ± 0.47 | 7.69 ± 1.25 | 0.01 | 0.093 ± 0.003 |
| | | 24.76 ± 0.47 | 7.61 ± 0.36 | 7.71 ± 1.62 | 0.05 | 0.043 ± 0.002 |
| | | 25.18 ± 0.52 | 7.59 ± 0.44 | 7.74 ± 1.58 | 0.1 | 0.092 ± 0.003 |

ND, not detected.

**Table 2.** Acute toxicity of $Hg^{2+}$ to rare minnow at different life stages.

| Stages | Time (h) | $LC_{50}$ (mg/L) | 95% Confidence Limit (mg/L) | Regression Equation |
|---|---|---|---|---|
| Embryos | 24 | 0.86 | 0.75–1.05 | Y = −1.88 + 2.18X |
| | 48 | 0.83 | 0.72–1.00 | Y = −1.77 + 2.13X |
| | 72 | 0.72 | 0.64–0.84 | Y = −1.67 + 2.30X |
| | 96 | 0.56 | 0.51–0.62 | Y = −1.82 + 3.25X |
| Newly hatched larvae | 24 | 0.16 | 0.14–0.22 | Y = −3.36 + 21.42X |
| | 48 | 0.12 | 0.11–0.15 | Y = −2.13 + 17.18X |
| | 72 | 0.10 | 0.08–0.11 | Y = −1.65 + 17.36X |
| | 96 | 0.07 | 0.06–0.08 | Y = −1.90 + 25.97X |
| Juvenile fish | 24 | 0.14 | 0.12–0.31 | Y = −3.71 + 27.07X |
| | 48 | 0.12 | 0.11–0.15 | Y = −4.02 + 33.17X |
| | 72 | 0.11 | 0.11–0.13 | Y = −2.19 + 18.50X |
| | 96 | 0.10 | 0.09–0.11 | Y = −2.16 + 21.63X |

*3.3. Sub-Chronic Toxicity of $Hg^{2+}$ to Rare Minnow*

The cumulative mortality increases with the extension of exposure time and the increase in concentrations, indicating that $Hg^{2+}$ can cause embryo-larval developmental toxicity in a dose-dependent and time-dependent manner (Table 3). A striking lethal effect on the embryos and larvae was significantly induced at no less than 0.05 mg/L or a higher concentration at 72–144 h of exposure ($p < 0.05$). The cumulative mortality of the 0.05 mg/L group was only approximately 8% at 96 h of exposure but reached 20% at 144 h of exposure. Compared with the control, the 0.10 mg/L group at 72 h presented significantly higher cumulative mortality ($p < 0.05$) and a more pronounced lethal effect over time. Although embryo and sac-fry stages are critical periods for oviparous fish, no studies have reported the developmental toxicity of $Hg^{2+}$ to the two stages in rare minnow. Our results showed that cumulative mortality surged from 8.33% in the 0.05 mg/L $Hg^{2+}$ group to 88.33% in the 0.1 mg/L $Hg^{2+}$ group at 96 h of exposure, suggesting that newly hatched larvae were more susceptible to $Hg^{2+}$ than embryos. Our findings were consistent with the acute toxicity of $Hg^{2+}$ to rare minnow at different life stages obtained above.

**Table 3.** Cumulative mortality of rare minnow at embryo and sac-fry stages after exposure to different concentrations of $Hg^{2+}$.

| Concentration (mg/L) | Time (hpf) | | | | | | |
|---|---|---|---|---|---|---|---|
| | 24 | 48 | 72 | 96 | 120 | 144 | 168 |
| 0.00 | $0.00 \pm 0.00$ | $0.00 \pm 0.00$ | $0.00 \pm 0.00$ | $0.00 \pm 0.00$ | $0.00 \pm 0.00$ | $0.00 \pm 0.00$ | $0.00 \pm 0.00$ |
| 0.001 | $0.00 \pm 0.00$ | $0.00 \pm 0.00$ | $1.67 \pm 2.89$ | $3.33 \pm 2.89$ | $5.00 \pm 0.00$ | $5.00 \pm 0.00$ | $5.00 \pm 0.00$ |
| 0.005 | $0.00 \pm 0.00$ | $0.00 \pm 0.00$ | $1.67 \pm 2.89$ | $5.00 \pm 0.00$ | $5.00 \pm 0.00$ | $5.00 \pm 0.00$ | $5.00 \pm 0.00$ |
| 0.01 | $3.33 \pm 2.89$ | $3.33 \pm 2.89$ | $5.00 \pm 5.00$ | $6.67 \pm 3.33$ | $6.67 \pm 3.33$ | $10.00 \pm 5.00$ | $10.00 \pm 5.00$ |
| 0.05 | $3.33 \pm 2.89$ | $3.33 \pm 2.89$ | $3.33 \pm 2.89$ | $8.33 \pm 5.77$ * | $10.00 \pm 8.66$ * | $16.67 \pm 7.64$ * | $20.00 \pm 13.23$ * |
| 0.10 | $3.33 \pm 2.89$ | $5.00 \pm 5.00$ | $51.67 \pm 12.58$ * | $88.33 \pm 2.89$ * | $98.33 \pm 2.89$ * | $100.00 \pm 0.00$ * | $100.00 \pm 0.00$ * |

*hph* h post-hatching, * $p < 0.05$.

$Hg^{2+}$ could also affect the hatching rate of embryos (Table 4). In the control group, embryos began hatching at approximately 72 hpf, while almost all surviving embryos hatched at 96 hpf. The hatching rate of embryos exposed to the 0.10 mg/L group was significantly accelerated at 72 hpf ($p < 0.05$) but was only 21 and 23% at 84 and 96 hpf, respectively, significantly inhibited compared to the control group ($p < 0.05$).

**Table 4.** Cumulative hatching rate of embryos after exposure to different concentrations of $Hg^{2+}$.

| Concentration (mg/L) | Time (hpf) | | |
|---|---|---|---|
| | 72 | 84 | 96 |
| 0.00 | $3.33 \pm 5.77$ | $63.33 \pm 7.64$ | $100.00 \pm 0.00$ |
| 0.001 | $0.00 \pm 0.00$ | $80.00 \pm 10.00$ | $98.33 \pm 2.89$ |
| 0.005 | $5.00 \pm 5.00$ | $60.00 \pm 8.66$ | $100.00 \pm 0.00$ |
| 0.01 | $0.00 \pm 0.00$ | $63.33 \pm 7.56$ | $93.33 \pm 7.64$ |
| 0.05 | $5.00 \pm 5.00$ | $88.33 \pm 10.41$ | $93.33 \pm 7.64$ |
| 0.10 | $16.67 \pm 10.41$ * | $21.67 \pm 10.41$ * | $23.33 \pm 7.64$ * |

*hph* h post-hatching, * $p < 0.05$.

$Hg^{2+}$ could induce sub-lethal effects in rare minnow at embryo and sac-fry stages, including heartbeat, malformation, and body length (Figure 1). No effect could be found on the spontaneous movements of rare minnow embryos at 36 hpf at all $Hg^{2+}$ exposure concentrations ($p > 0.05$; Figure 1A). Compared with that of the control, the heart rate of rare minnow embryos at 48 hpf was significantly inhibited at no less than 0.05 mg/L of $Hg^{2+}$ ($p < 0.05$) (Figure 1B). The heart rate of rare minnow embryos exposed to 0.05 mg/L $Hg^{2+}$ was inhibited by 15%. Moreover, the inhibition rate was nearly 22% at 0.10 mg/L $Hg^{2+}$. During the short-term toxicity test of $Hg^{2+}$ to rare minnow at embryo and sac-fry stages, developmental abnormalities were observed (Figure 2), including pericardial edema (PE), yolk sac edema (YSE), a bent spine (BS), a bent tail (BT), yolk sac necrosis and an uninflated swim bladder (USB). The malformation rate increased with the concentrations, indicating that $Hg^{2+}$ could cause embryo-larval developmental malformation in a dose-dependent manner. The malformation rate in groups of high concentrations (0.05 and 0.10 mg/L) was significantly induced and higher than 95% ($p < 0.05$; Figure 1C). It was worth noting that $Hg^{2+}$ induced growth retardation, and that the body length of larvae with $Hg^{2+}$ concentrations at more than 0.05 mg/L was significantly shorter than that of those in the control group ($p < 0.05$; Figure 1D).

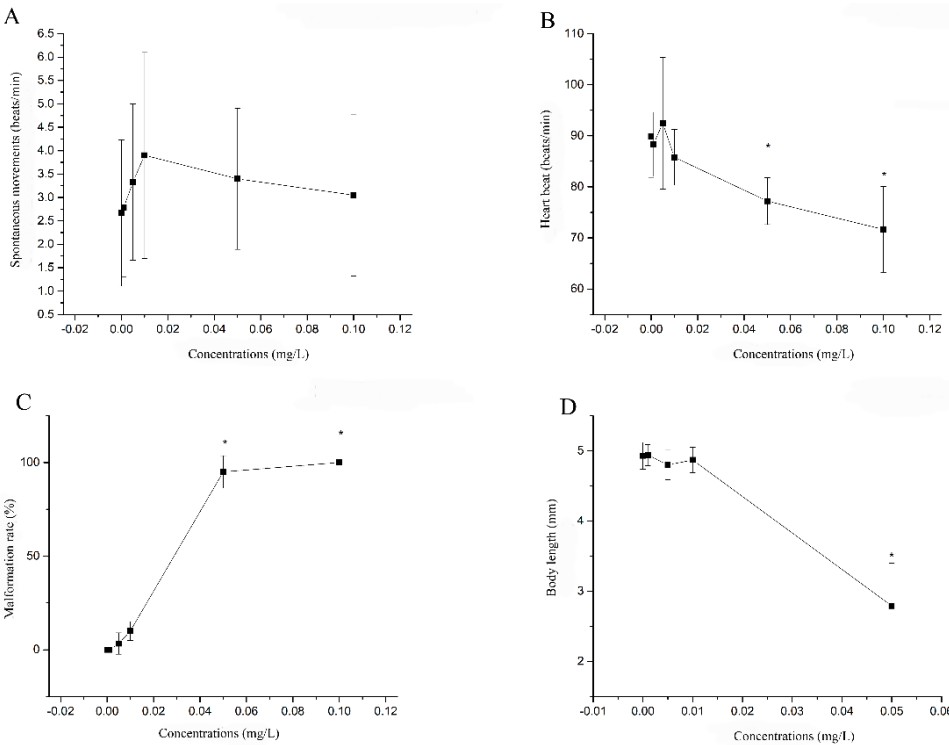

**Figure 1.** Sub−lethal effects of Hg$^{2+}$ on rare minnow embryo and sac−fry stages. (**A**) frequency of spontaneous movement at 36 hpf; (**B**) heart rate at 48 hpf; (**C**) malformation rate at 96 hpf; (**D**) body length of larvae after exposure. * Significant difference from the control group ($p < 0.05$).

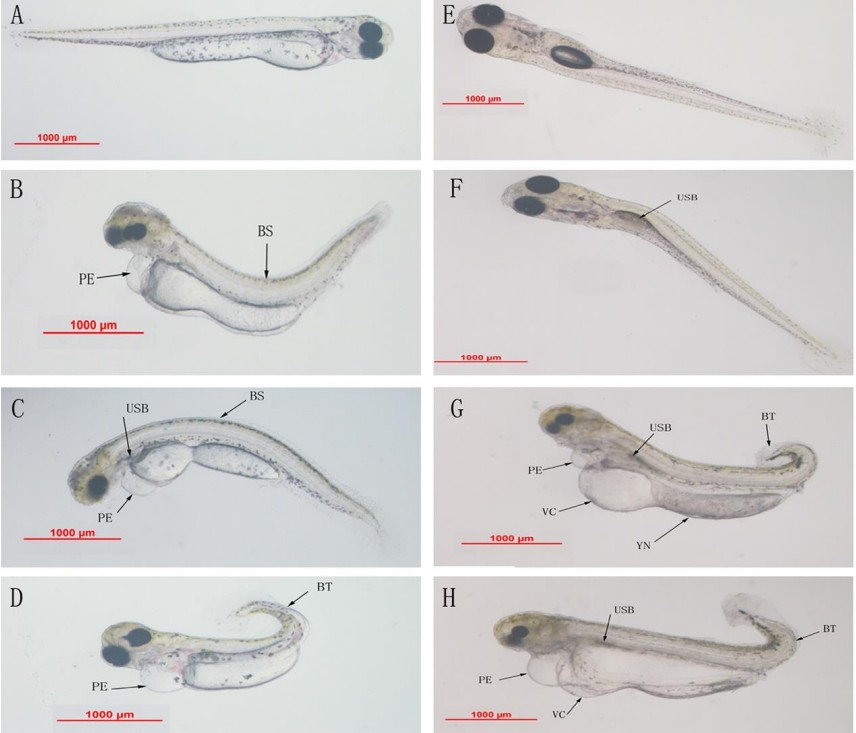

**Figure 2.** Microscopic images of newly hatched rare minnow larvae (**A–D**) and larvae at 3 d post hatch (**E–H**): (**A**) normal newly hatched larva, (**B–D**) Hg$^{2+}$ exposed newly hatched larva with pericardial edema (PE), bent spine (BS) and bent tail (BT), (**E**) normal larva at 3 d post hatch, (**F–H**) Hg$^{2+}$-exposed larva with uninflated swim bladder (USB), pericardial edema (PE), yolk sac edema (YSE), bent spine (BS), bent tail (BT) and yolk sac necrosis.

According to the results, $Hg^{2+}$ exerted no effects on the spontaneous movements at no more than 0.10 mg/L; the LOEC and NOEC values of $Hg^{2+}$ to hatching rate at 96 hpf were 0.10 and 0.05 mg/L, respectively. The LOEC and NOEC values of $Hg^{2+}$ to heartbeat, mortality, malformation rate, and body length of the survived larvae were 0.05 and 0.01 mg/L, respectively (Table 5).

**Table 5.** LOEC and NOEC values of $Hg^{2+}$ to rare minnow at embryo and sac-fry stages.

| Endpoints | LOEC (mg/L) | NOEC (mg/L) |
| --- | --- | --- |
| Spontaneous movements (36 hpf) | >0.1 | >0.1 |
| Hatching rate (96 hpf) | 0.1 | 0.05 |
| Heartbeat (48 hpf) | 0.05 | 0.01 |
| Mortality (96 hpf) | 0.05 | 0.01 |
| Malformation rate (96 hpf) | 0.05 | 0.01 |
| Body length | 0.05 | 0.01 |

## 4. Discussion

Numerous fish were studied to investigate the toxicity of inorganic mercury. Our experiments showed that $Hg^{2+}$ had acute toxicity to rare minnow at different life stages, further confirming the conclusion of Li et al. [23] that $Hg^{2+}$ showed acute toxicity to rare minnow larvae. Ecotoxicology data in this study provide important guidance to establish WQS to protect organisms in the aquatic environment, which is obtained with a protection factor of 100 ($LC_{50}/100$). Based on present data, a safe level of $Hg^{2+}$ exposure for rare minnows was 0.7 µg/L by determining the most sensitive stage (i.e., newly hatched larvae), higher than the WQS for fisheries (GB 11607-89; 0.5 µg/L) proposed by the Ministry of Agriculture of China. This indicated that both permissible limits were appropriate for the hatchery and nursery of rare minnow.

Research on the sensitive period was of great importance for protecting the organisms from external toxic harm. Our study showed that the sensitivity to $Hg^{2+}$ toxicity of rare minnow was dramatically different at various stages. The newly hatched larvae were the most sensitive, approximately two to ten times the sensitivity of juvenile fish and embryos, respectively. Similar results were reported by Roos-Muñoz et al. [26], who found that larvae were more sensitive to $Hg^{2+}$ compared to juvenile fish (96 h $LC_{50}$ 0.25 mg/L vs. 0.50 mg/L). On the other hand, fish embryos were reported to be more sensitive than larvae and juvenile fish in another study [27]. Firstly, embryos were more resistant than newly hatched larvae and juvenile fish; we confidently attribute this to the protective effect of the chorion against $Hg^{2+}$ [28] or the detoxification route through gills transport [29]. Chang et al. [30] described this process as not yet functional within 96 hpf, so only larvae and juvenile fish were able to take advantage of this physiological process. Secondly, newly hatched larvae were more sensitive than juvenile fish, because their organs (such as the liver) had not yet been fully developed. In contrast, juvenile fish could produce metallothioneins [31]. Thirdly, juveniles and larvae presented a clear difference in the ratio of body volume to body surface [32]. Smaller individuals have a greater ratio of surface area to body weight, and thus may accumulate $Hg^{2+}$ relatively quickly and die sooner. However, fish sensitivity at different life stages still remains to be determined in future research.

Hatching was a main toxicological endpoint for fish embryos. In this study, the rare minnow embryos showed a decreased hatching success rate at concentrations ≥100 µg/L $Hg^{2+}$. Similar findings were reported in other fish; for instance, Huang et al. [33] found the flounder embryos showed a decreased hatching success rate at concentrations ≥20 µg/L $Hg^{2+}$. $Hg^{2+}$ could remarkably reduce the hatching success rate of zebrafish at 16 µg/L [34] and was completely inhibited at 32 µg/L [28]. The toxic mechanisms of $Hg^{2+-}$ exposed embryos may be due to structural and functional disturbances during embryonic development [35]. It was noticed that the hatching rate of rare minnow embryos decreased after being exposed to $Hg^{2+}$, while developmental abnormalities increased. Further study

is necessary to elucidate the relationship between developmental abnormalities and the hatching rate.

Developmental abnormalities play a vital role in evaluating the teratogenicity of heavy metals [35]. In the study, malformed larvae exhibiting PE, YSE, BS, BT and yolk sac necrosis were induced by $HgCl_2$ exposure, consistent with those reported in Japanese flounder (*Paralichthys olivaceus*) [33] and medaka (*Oryzias latipes*) [36]. A sensitive endpoint plays a predominant role in the ecotoxicological evaluation of chemical pollutants. With a lower LOEC value, the toxicological endpoint is considered to be more sensitive. Based on the results, the LOEC values of $Hg^{2+}$ to mortality, heartbeat, malformation rate and body length of the surviving larvae were lower than those of the spontaneous movement and hatching rate, suggesting mortality, heartbeat, malformation rate and body length could serve as sensitive endpoints to assess the hazard of $Hg^{2+}$ to fish at early life stages. The consistent LOEC values of $Hg^{2+}$ to mortality, heartbeat, malformation rate and body length of the survived larvae revealed that the sensitivity of body length was similar to that of the other three factors, consistent with the result found by Luo et al. [24]. Therefore, we suggested that the evaluation of body length should be included in the rare minnow short-term toxicity test on embryo and sac-fry stages as a necessary routine test indicator.

## 5. Conclusions

The acute toxicity results showed that the newly hatched larvae were the most sensitive, followed by juvenile fish, while the embryo stage was the most resistant period to $Hg^{2+}$ exposure. The generated data of this study indicated that a safe level of $Hg^{2+}$ exposure for rare minnow should be 0.7 μg/L. In addition, the short-term developmental toxicity results revealed that the heartbeat, embryo hatching rate, and body length of surviving larvae could be chosen as sensitive endpoints for assessing the short-term developmental toxicity of $Hg^{2+}$ to rare minnow.

**Author Contributions:** Conceptualization, X.X., Y.Z. (Yu Zeng) and Y.H.; methodology, Q.S. and H.L. (Hao Liu); software, Q.S.; investigation, Q.Z.; resources, J.W.; data curation, H.L. (Huatao Li) and P.H.; writing—original draft preparation, X.X. (Xiaoqin Xiong); writing—review and editing, Z.W.; supervision, Y.Z. (Yuanchao Zou); project administration, Y.Z. (Yu Zeng) and Y.H.; funding acquisition, X.X., Y.Z. (Yuanchao Zou) and Y.H. All authors have read and agreed to the published version of the manuscript.

**Funding:** This work was funded by the Projects of Sichuan Provincial Department of Science and Technology (NO. 2021YFS0359, NO. 2021YFN0033, NO. 2022NSFSC0129), the Foundation of Ph.D. Scientific Research of Neijiang Normal University (NO. 18B13), the research project of Neijiang Normal University (NO. Z2019068), the Natural Science Foundation of Hebei Province (No. C2019204360), the Science and Technology Project of Hebei Education Department (QN2020132).

**Institutional Review Board Statement:** Not applicable.

**Informed Consent Statement:** Not applicable.

**Data Availability Statement:** Not applicable.

**Acknowledgments:** We would like to express our gratitude and appreciation to those who have taken the time to critically review this article.

**Conflicts of Interest:** The authors declare no conflict of interest.

## Abbreviations

Hg—Mercury; LOEC—Lowest Observed Effect Concentration; NOEC—No Observed Effect Concentration; WQC—Water Quality Criteria; CV-AAS—Cold Vapor Atomic Absorption Spectrometry.

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
