# Peer review of "Assessment of Acute and Short-Term Developmental Toxicity of Mercury Chloride to Rare Minnow (Gobiocypris rarus)"

_water, doi:10.3390/w14182825_

Round 1

Reviewer 1 Report

The paper of Xiong et al presented the effects of acute toxicity of mercury to different developmental stages of rare minnow.  Although this paper doesn't have any novelty, I think it is worth publishing as it adds existing knowledge in terms of toxicological assessment as well as the roles of biomarkers in ecotoxicological assessment. I have minor comments and suggestions. Please see the attached manuscript.

Author Response

Thank you very much for offering us the opportunity to revise the manuscript entitled “Assessment of acute and short-term developmental toxicity of mercury chloride to rare minnow (Gobiocypris rarus)” (ID water-1844534). We appreciate the time and effort that you and the reviewers dedicated to providing feedback on our manuscript and are grateful for the insightful comments on our paper. We have studied the comments carefully and made corrections which we hope can meet with your approval. The main corrections are marked in the revised manuscript and the responds to the reviewers’ comments are as follows.

Reviewer 1: The paper of Xiong et al presented the effects of acute toxicity of mercury to different developmental stages of rare minnow.  Although this paper doesn't have any novelty, I think it is worth publishing as it adds existing knowledge in terms of toxicological assessment as well as the roles of biomarkers in ecotoxicological assessment. I have minor comments and suggestions. Please see the attached manuscript.

1.     Before this sentence please insert the importance of biomarkers and cite the works (p.2 Lines 46-47).

Reply: We thank the reviewer for pointing this out, and have revised. (Please see p.2 lines 49-50 in the revised manuscript.)

2.     How about other information in 3.1. Chemical analysis, such as Water temperature, pH, dissolved oxygen? The authors must also disclose the data.

Reply: We thank the reviewer for reminding the valuable suggestions, and have supplemented relevant data in the revised manuscript. (Please see Table 1) 

Reviewer 2 Report

The present study reports the acute and short-term developmental toxicity of mercury chloride to rare minnow (Gobiocypris rarus). The result seems interesting, however some points remain to be revised or clarified before acceptance.

Introduction

The relevant information should be provided in this section. For example:

1.      What’s the environmentally relevant concentration of Hg2+ in aquatic environment ? Please add this information.

2.      What’s the toxic effect of Hg2+ to aquatic organisms? Please add this information.

3.      Line 39: The authors mentioned that “However, the toxicity of Hg to fish nearly remains unknown.” However, the similar work has been reported by several studies on toxicity of Hg2+ on fish.

Materials and methods

1.      Lines 91-92: please modify this sentence.

2.      Line 46: modify this sentence.

3.      Line 113 and Figure 1A: what’s the spontaneous movement?

Results and Discussion

1.      Table 2: For the newly hatched larvae, why did the authors determine the LC50 of 4 hours instead of 24 hours?

2.      Table 3: What’s meaning of “Coasts”?

3.      Table 4: The hatching rate of embryos exposed to the highest concentration of Hg2+ (0.10 mg/L) was significantly accelerated at 72 hpf. Why?

4.      Line 256-257: please provide the Latin names of the two fish species.

A few grammatical mistakes were found, please rectify before your article gets published.

Author Response

Thank you very much for offering us the opportunity to revise the manuscript entitled “Assessment of acute and short-term developmental toxicity of mercury chloride to rare minnow (Gobiocypris rarus)” (ID water-1844534). We appreciate the time and effort that you and the reviewers dedicated to providing feedback on our manuscript and are grateful for the insightful comments on our paper. We have studied the comments carefully and made corrections which we hope can meet with your approval. The main corrections are marked in the revised manuscript and the responds to the reviewers’ comments are as follows.

Reviewer #2: The present study reports the acute and short-term developmental toxicity of mercury chloride to rare minnow (Gobiocypris rarus). The result seems interesting, however some points remain to be revised or clarified before acceptance.

Introduction

The relevant information should be provided in this section. For example:

1. What′s the environmentally relevant concentration of Hg2+ in aquatic environment? Please add this information.

Reply: We thank the reviewer for the important suggestions, and have added this data in the revised manuscript, Hg concentrations vary from 0.038 to 10.6 μg/L in China. (Please see p.2 lines 51-52)

2.  What′s the toxic effect of Hg2+ to aquatic organisms? Please add this information.

Reply: Thank you for your valuable suggestions. We have added the toxic effect of Hg2+ to aquatic organisms in the revised manuscrip. (Please see p.2 lines 52-53)

3. Line 39: The authors mentioned that “However, the toxicity of Hg to fish nearly remains unknown.” However, the similar work has been reported by several studies on toxicity of Hg2+ on fish.

Reply: Thank you for your patience review. We have modified it and present it in a more accurate way in line 42, which means that compared with the toxicity of Hg pollution on humans and wildlife, the toxicity of Hg to fish far less is still fully understand.

Materials and methods

1. Lines 91-92: please modify this sentence.Reply: Thank you for your valuable suggestions. We have modified it based on your comments. (Please see Line 97)2. Line 46: modify this sentence.

Reply: Thank you for your valuable suggestions. We have made a suitable modification based on your comments. (Please see Line 51)

3. Line 113 and Figure 1A: What′s the spontaneous movement?

Reply: The spontaneous movement refers to the number of the embryo can be activated without any external stimulation.

Results and Discussion

1. Table 2: For the newly hatched larvae, why did the authors determine the LC50 of 4 hours instead of 24 hours?

Reply: We confirm this is an important mistake, it is 24 hours, and we have modified in the revised version. (Please see Table 2)

2. Table 3: What′s meaning of “Coasts”?

Reply: Maybe it was a system upload error, “Coasts” was mistake, it was time, we have corrected it, please see Table 3.

3. Table 4: The hatching rate of embryos exposed to the highest concentration of Hg2+ (0.10 mg/L) was significantly accelerated at 72 hpf. Why?

Reply: Referring to correlative theories and literature search, the most possible reason is “Hormesis”, and similar phenomenon has also been found in other studies, for example zebrafish embryos[1], mallard eggs[2] and so on.[1] L.C. Hu.; X.W. Yu.; Z. K. Zhao.; J. L.Yuan.; X. Yang. Study on embryo toxicity of bensulfuron-methyl in zebrafish. Journal of Public Health and Preventive Medicine.2011, 22(6): 1-4.[2] Heinz, G. H.; Hoffman, D. J.; Klimstra, J. D.; Stebbins, K. R.; Kondrad, S. L.; Erwin, C. A. Hormesis associated with a low dose of methylmercury injected into mallard eggs. Archives of Environmental Contamination and Toxicology. 2012, 62(1), 141-4.

4. Line 256-257: please provide the Latin names of the two fish species.

Reply: Thank you for your valuable suggestions. We have provided it. (Please see line 267)

5. A few grammatical mistakes were found, please rectify before your article gets published.

Reply: Thank you for your valuable suggestions. As suggested by the reviewer, we have revised those grammatical mistakes in the revised version.

Round 2

Reviewer 2 Report

Accept